# How Fast Is Europe Getting Old? Analysis of Dynamics Applying the Spatial Shift–Share Approach

**Elżbieta Antczak**  **and Karolina Lewandowska-Gwarda** *

Faculty of Economics and Sociology, University of Lodz, Ul. Rewolucji 1905 r. 37, 90-214 Lodz, Poland;
elzbieta.antczak@uni.lodz.pl

\* Correspondence: karolina.lewandowska-gwarda@uni.lodz.pl; Tel.: +48-42-635-51-12

**Abstract:** In this article, we analyzed the dynamics of the population aging process in Europe. The study was conducted on the basis of statistical data on the number of people aged 65 and above per 1000 of the population in 32 European countries in the years 1991–2018. The analyses also took into account the structure of the population by gender in five age groups: 65–69, 70–74, 75–79, 80–84, and 85 and above. An extensive analysis of the rate of changes in the magnitude of the phenomenon was carried out, which gave an answer to the question about how quickly Europe is aging. We applied the spatial dynamic shift–share method. The spatial variant of the method allowed, among others, indicating countries where the pace of population aging in a specific age group was faster/slower than in locations neighboring the examined country. Specific regions characterized by the fastest population aging were also indicated, and shares of structural and sectoral factors of the changes were estimated. Furthermore, based on the values of local competitiveness indicators, regions were identified where the aging of the population decelerated or accelerated the phenomenon in neighboring countries in the study period.

**Keywords:** aging of the population; sustainable development; structural and geographical analysis; spatial relationships; spatiotemporal analysis; dynamics of changes; European countries

---

## 1. Introduction

Aging or greying of the population is a process characterized by the increasing share of older people (65 and over) in the population, which causes a permanent change in the age structure of the population. A United Nations report shows that this process is most advanced in Europe (the population pyramids that indicate changes in the national age structure for selected years and forecasts are presented in the Appendix A). The populations of other regions are growing older as well. In 2018, Northern America and Australia (16% of the population) were right behind Europe (20%) in the aging statistics, followed by Asia (11%), Latin America and the Caribbean (8%), and finally, Africa (4%) [1,2].

In 2018, the share of people aged 65 and over in the European Union population reached 19.7% [3]. According to forecasts presented by the European Commission, this will rise to 29% in 2070, while the share of the working-age population (aged 15–64) will decline from 64.7% to 56%. As a result of those trends, the demographic old-age dependency ratio (people aged 65 or above relative to the working-age population) is projected to increase from 30.5% to 51.2%. That implies that in 2070, there will be only around two working-age individuals per every person aged over 65 years. The main determinants of those processes are higher life expectancy at birth, which is forecast to rise from 81 in 2018 to 90.3 in 2070, and decreased fertility among European populations [4]. Such a prospect constitutes a serious challenge to countries, because it forces changes and adjustments in almost every sphere of state activity—labor market, demand for goods and services, urban planning, infrastructure

development, etc. Economies have to prepare for increasing fiscal costs linked to pensions, healthcare, and long-term care.

The Sustainable Development Goals, set by the United Nations General Assembly in 2015, were developed to secure a better and more sustainable future for all people in the world. Aging of the population has strong implications for the sustainable development and implementation of its assumptions. Almost every goal requires taking actions related to older people. For example, to achieve Goal 1 (No poverty), older people should be prevented from falling into poverty, which requires flexible retirement policies, adjustments that could help them to stay longer on the labor market, and providing social assistance for poor older people and their families. Goal 3 (Good health and well-being) requires transformation of health and social systems from a disease-only focus towards the provision of integrated and person-centered care also in terms of social needs. It is extremely important to reach assumptions of Goal 5 (Gender equality), as women are still discriminated against in many aspects of socioeconomic life. Generally, women live longer than men; they accounted for 56.9% of the European Union population aged 65 and above in 2018 [3]. Although women actively participate in the labor market, provide child-care and long-term care for the elderly, they experience wage discrimination, which has many negative consequences in later life, including a greater risk of poverty (reduced access to pensions) and poorer access to health and social care services. Therefore, there is a need to promote equitable remuneration for men and women, to raise the status of older women within households, and to improve access to all kinds of services, from medical to social and economic ones. Goal 9 (Industry, innovation, and infrastructure) indicates a strong need to foster innovations that target changes brought about by population aging. The development of e-Health technology is very important for all patients, but in the case of older people, it can provide, for example, health monitoring that can help to maintain independence, avoid hospitalization, and improve quality of life by ensuring a sense of security. One of the most crucial is the last one—Goal 16 (Peace, justice, and strong institutions). To achieve it, the world should understand the process of aging and all problems that go therewith. Old people are often stereotypically perceived as weaker and burdensome; the society does not appreciate or even rejects them. Prejudice and discrimination against older people create barriers to social, economic, and political development. For that reason, societies should be made aware of and accustomed to old age; they ought to appreciate older people and use their potential. Tackling ageism and older people's invisibility will empower them to achieve things that previous generations could never imagine [5].

As the European society aging process affects countries' socioeconomic condition and causes changes in almost every sphere of state activity, the monitoring and analysis of demographic changes is a priority task today. Research into that area is extremely important because it enables responding to the needs of the changing society and effects of the aging process. Therefore, the main aim of the article was to analyze the dynamics of the aging process in Europe. The study was conducted using the dynamic spatial shift–share analysis method (SSSA). Thus, an extensive and long-term analysis of the rate of changes in the magnitude of the phenomenon was carried out, which allowed answering the question about how quickly Europe is aging. Specific countries characterized by the fastest dynamics of population aging were also indicated, and the shares of structural and sectoral factors of the changes were estimated. Additionally, the study considered interregional relationships in the form of a weights matrix based on geographical distance. The spatial variant of the shift–share method allowed, among others, indicating countries where the pace of population aging in a particular age group was faster (slower) than in locations neighboring the country. Moreover, based on the values of local competitiveness indicators, regions were identified where the aging of the population decelerated or accelerated the phenomenon in neighboring countries in the study period. The analyses were carried out on the basis of statistical data obtained from Eurostat regarding the number of people aged 65 and above per 1000 of the population in the years 1991–2018. The study was not performed merely for the European Union countries: In order to provide the most complete picture of the European society aged 65 and above, the perspective was extended to include Iceland, Liechtenstein, Norway,

and Switzerland. Thus, analyses were conducted for 32 European countries. The study also took into account the population structure by gender in five age groups: 65–69, 70–74, 75–79, 80–84, and 85 and above.

The study consists of five parts. Section 2 describes main issues raised in the literature on population aging and indicates the contribution of this article to the literature. Section 3 presents a databank used in the study and results of a preliminary data analysis using GIS and basic statistics. It also describes the method applied in the further analysis—the spatial dynamic shift–share method. Section 4 discusses the results of the European population aging analyses in detail. The final section provides general conclusions and indicates further directions of research.

## 2. Population Aging in Literature

The issue of the population aging and its impact on socioeconomic development has been widely discussed in the literature. Many authors also analyzed the causes of the process. They indicated that a demographic reason for population aging was baby boom observed in many industrialized countries from the end of the Second World War to the mid-sixties of the twentieth century [6]. Political stabilization and economic development provided favorable conditions to start families at that time. Unfortunately, the postwar generation was not eager to have as many children as their parents; hence, they constitute a larger generation than the previous and subsequent ones. Nevertheless, the fact that they are currently attaining retirement age would not be enough to start the population aging process. An important factor is that, in general, people live longer thanks to significant advances in healthcare (vaccinations, blood chemistry analysis, medicines, new medical technologies, greater access to healthcare, etc.) and better standards of living [7,8]. Moreover, the fertility rate is dropping in many countries. The reasons for that are complex and certainly vary from country to country, but perhaps the most crucial one is the broadly understood socioeconomic development. Countries where women have options of education and employment, where people's economic status is higher, and where more people live in cities than in the countryside are generally characterized by lower birth rates (the modern family model is usually 2 + 1) and older age of having the first child [9]. Migrations are another factor that can affect the age structure of the population. Highly developed European countries that attract foreigners due to higher living standards and good working conditions can benefit from large immigration flows and slow the aging process when they implement supportive migration policies. However, the literature emphasizes that countries are not able to control immigration to provide a steady flow of people of working age; therefore, migration cannot be the only remedy for an aging population. At the same time, many countries, especially in Eastern Europe, experienced the emigration of young people after joining the European Union and opening labor markets, which is a very negative phenomenon accelerating the population aging process [1].

The population aging process requires not only constant monitoring, but also population analyses and forecasts that allow preparing for changes. Every three years, the Ageing Working Group of the Economic Policy Committee and the European Commission's Directorate-General for Economic and Financial Affairs publish the Ageing Report presenting an analysis of the impact of aging populations on the labor market and potential economic growth. The latest, 2018 report widely describes Eurostat's population projections up to 2070 and age-related expenditures for all European Union members. The report clearly shows that fiscal costs linked to pensions, healthcare, and long-term care are expected to rise over the coming decades, as Europe's population continues to age significantly [4]. Therefore, a common question that appears in literature is "Can Europe afford to grow old?" Researchers indicate that population aging will have major repercussions for European labor markets, economic growth, and public finances [10–12]. However, the aging of the society is a slow process, which is why countries and economies can prepare for it and take steps to mitigate its adverse consequences. Carone and Costello pointed out that EU member states' policies need to adapt to the new reality. The key issue is to design pension systems that would be sustainable in the face of uncertain economic and demographic developments (an additional reform to raise the average age of leaving the labor market would be

needed to stabilize the adult life portion spent in retirement). An even more important and complex challenge is the adjustment of the healthcare and long-term care systems, which requires investment in medical research and new technologies [10]. Those tasks seem to be impossible to achieve as they involve profound fiscal consequences. Thus, the sustainability of public finances has become a part of everyday political debate [12]. Nevertheless, population aging not only brings about problems and challenges, but also creates new opportunities. One of those is silver economy, which is defined by the European Commission as "economic opportunities brought by expenditure related to population ageing and the needs of citizens over the age of 50" [13]. The needs of the aging population concern not only healthcare, but also other products and services such as food, services related to housing, sports and health clubs, tourism, wellness, cultural institutions, and education. Therefore, it creates impulses prompting entrepreneurship and new employment opportunities, which can positively influence economic growth [14].

The specialist world literature offers a lot of research on the depopulation and population aging applying statistical and econometric methods, including the shift–share analysis (SSA) and its various modifications. For example, Perry and Hayward used that to assess the impact of population changes in different age groups on the size of population migration among New Zealand regions [15]. Jones conducted a comprehensive analysis of changes in the population age structure as a determinant of labor productivity in countries of the world using the shift–share method [16]. In turn, Davis and Rodriguez employed the classical SSA to analyze the dynamics of the structure of employees aged 55 and above for the US states [17]. A similar study for Portuguese regions was carried out by Albuquerque and Ferreira [18]. In Poland, a dynamic approach to share–shift analysis was utilized to assess the population aging rate at the level of subregions [19]. Nevertheless, there is a lack of studies analyzing in detail the problem of population aging in Europe simultaneously taking into account different cross-sections. Therefore, this study contributes to the literature by using the spatial dynamic shift–share analysis method, which enabled a thorough analysis of population aging in the long run (1991–2018) for women and men in 32 European countries, in five age categories of elderly age. As a result of this study, not only a picture of changes taking place in Europe, but also detailed information on the changes occurring in individual countries and other cross-sections were obtained. A study taking such a comprehensive approach has never been carried out before and should be relevant because it enables understanding the society aging process impacting strongly on socioeconomic development.

## 3. Data and Methodology

### 3.1. Preliminary Data Analysis

The analysis of the aging process dynamics in Europe was carried out on the basis of statistical data obtained from Eurostat on the number of inhabitants per 1000 of the population in several cross-sections: age, gender, and space in the years 1991–2018. The research was conducted for five older age categories: 65–69, 70–74, 75–79, 80–84, and 85 and above, and 32 European countries. In addition to countries belonging to the European Union, i.e., Austria (AT), Belgium (BE), Bulgaria (BG), Croatia (HR), Cyprus (CY), the Czech Republic (CZ), Denmark (DK), Estonia (EE), Finland (FI), France (FR), Germany (DE), Greece (EL), Hungary (HU), Ireland (IE), Italy (IT), Latvia (LV), Lithuania (LT), Luxembourg (LU), Malta (MT), the Netherlands (NL), Poland (PL), Portugal (PT), Romania (RO), Slovakia (SK), Slovenia (SI), Spain (ES), Sweden (SE), and the United Kingdom (UK), the analysis included Iceland (IS), Lichtenstein (LI), Norway (NO), and Switzerland (CH).

The total population aged 65 and above per 1000 of residents in the analyzed countries was characterized by a steady increase over the study period (from 130 in 1991 to 189 in 2018, respectively). However, two stages were clearly marked. The first stage, until 2010, in which the variable showed slower dynamics—an increase of fewer than 2 people from year to year—and the second stage, after

2010, where the acceleration of the process could be observed—an average annual growth of about 4 people, Figure 1.

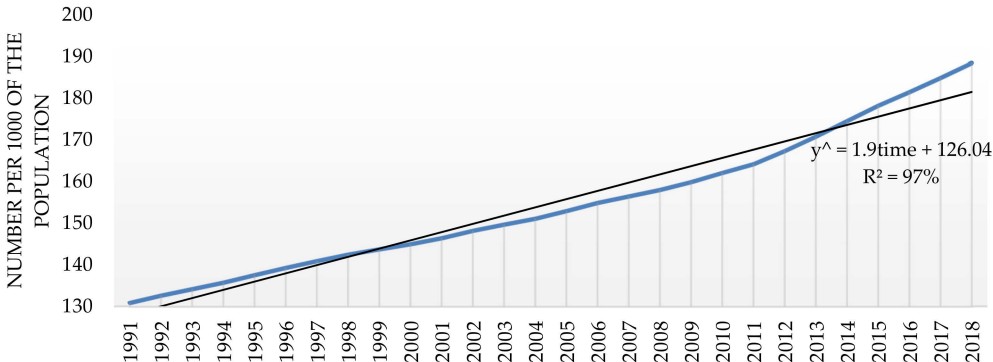

**Figure 1.** Total number of people aged 65 and above per 1000 of the population in 32 European countries in the years 1991–2018.

Women clearly predominated among individuals aged 65 and above. The difference between the number of women and men remained at the level of 29 in the years 1991–2018. Both variables were characterized by steady growth. The number of women aged 65 and above per 1000 of the population increased from 79 in 1991 to 108 in 2018, while the number of men from 51 to 80 respectively, Figure 2.

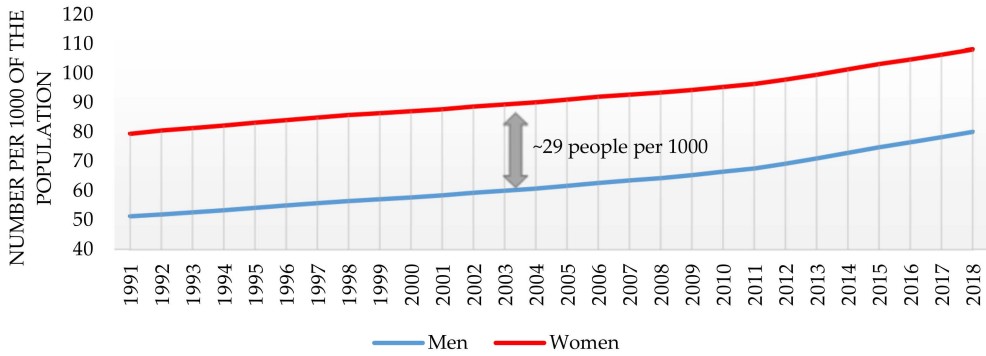

**Figure 2.** The total number of women and men aged 65 and above per 1000 of the population in 32 European countries in the years 1991–2018.

Throughout the analyzed period, the largest group was people aged 65–69. The variable ran at about 45 people per 1000 of the population until 2011; there was a marked increase of about 2 people from year to year in the years 2012–2016. In 2018, the number of people in that age category was 57 per 1000 of the population. For individuals aged 70–74, there were two considerable rises in the variable level: in 1992–1995 and 2017–2018. It should be emphasized that it was in that age category that the highest population growth per 1000 of inhabitants was observed in the analyzed period: from 16 people originally, reaching 46 in 2018. A decrease in the population aged 75–79 per 1000 of residents was noted in 1992–1995. A steady growth in subsequent years meant that the variable reached the level of 36 people in 2018. As for individuals aged 80–84, a drop per 1000 of the population was recorded in 1996–2000, whereupon the level of the variable increased to as many as 26 in 2018. The largest increase in the analyzed period was recorded for the oldest people—aged 85 and above, Figure 3.

The maps in Figure 4 clearly show that there was an increase in the population of senior citizens in all the countries in the years 1991–2018. In addition, the number of people aged 65 and above was characterized by quite significant spatial diversity. A clear division of continental Europe into the older western part and the younger eastern part can easily be seen. In 1991, the highest values of the variable were recorded in Northern Europe: in Sweden and Norway. In 2000, Germany and Belgium, as well as

countries of southern Europe—Portugal, Spain, Italy, Greece, and Bulgaria—joined that group. In 2010, Germany and Italy were characterized by the largest number of older people. In 2018, over half of the analyzed countries already belonged to that group. The lowest values of the variable were recorded in Ireland, Iceland, and Slovakia throughout the analyzed period.

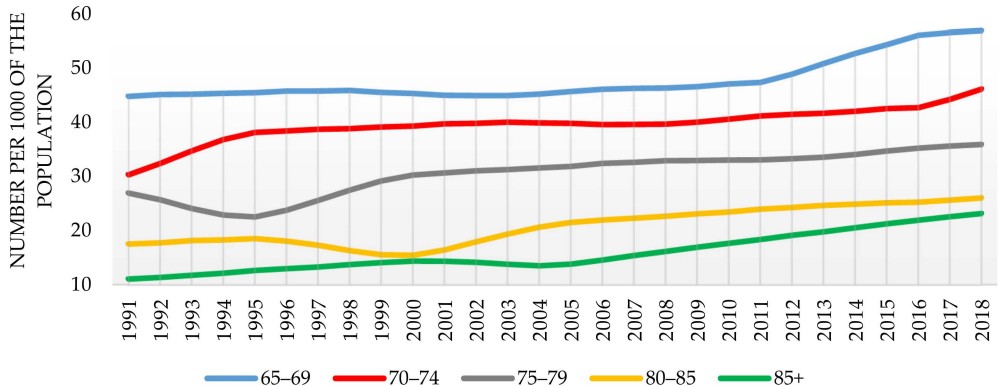

**Figure 3.** Total number of people per 1000 of the population in particular age categories in 32 European countries in the years 1991–2018.

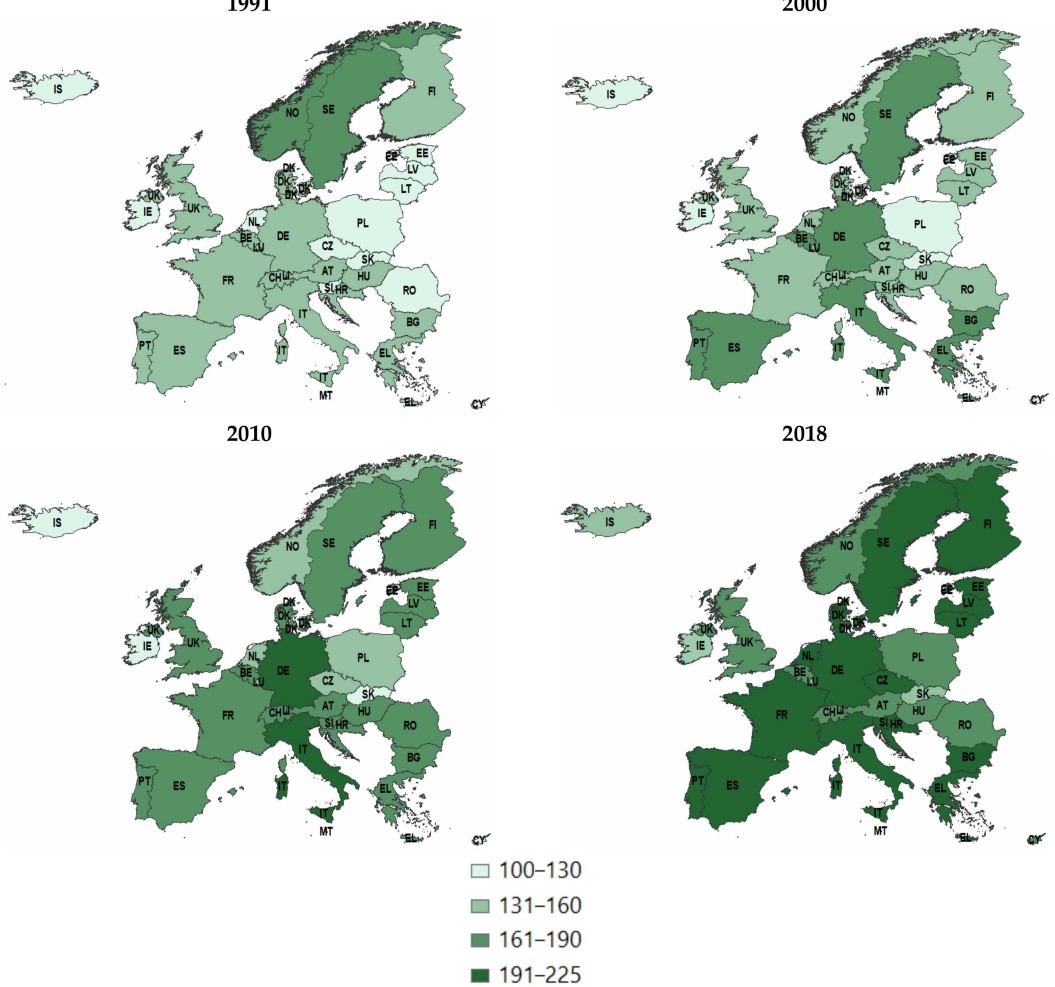

**Figure 4.** Number of people aged 65 and above per 1000 of the population in European countries in selected years.

As previously noted (Figure 2), women clearly predominated among older people in the entire study period. The biggest difference (of 68 people) between the number of women and men aged 65 and above per 1000 of the population was recorded in Latvia, while the smallest in Iceland (of 6 people). It can be clearly seen on the maps in Figure 5 that the number of women aged 65 and above per 1000 of the population was more spatially diversified than the number of men in Europe. The lowest values of the variable for men characterized the eastern part of Europe, as well as Ireland and Iceland, whereas the highest were reported for Sweden, Germany, Italy, and western continental Europe. As for women, over the years, a significant increase in the number of senior citizens per 1000 of the population was noted in the eastern part of the continent, whilst the countries with the lowest numbers throughout the research period included Ireland, Iceland, Cyprus, Liechtenstein, and Slovakia, Figure 5.

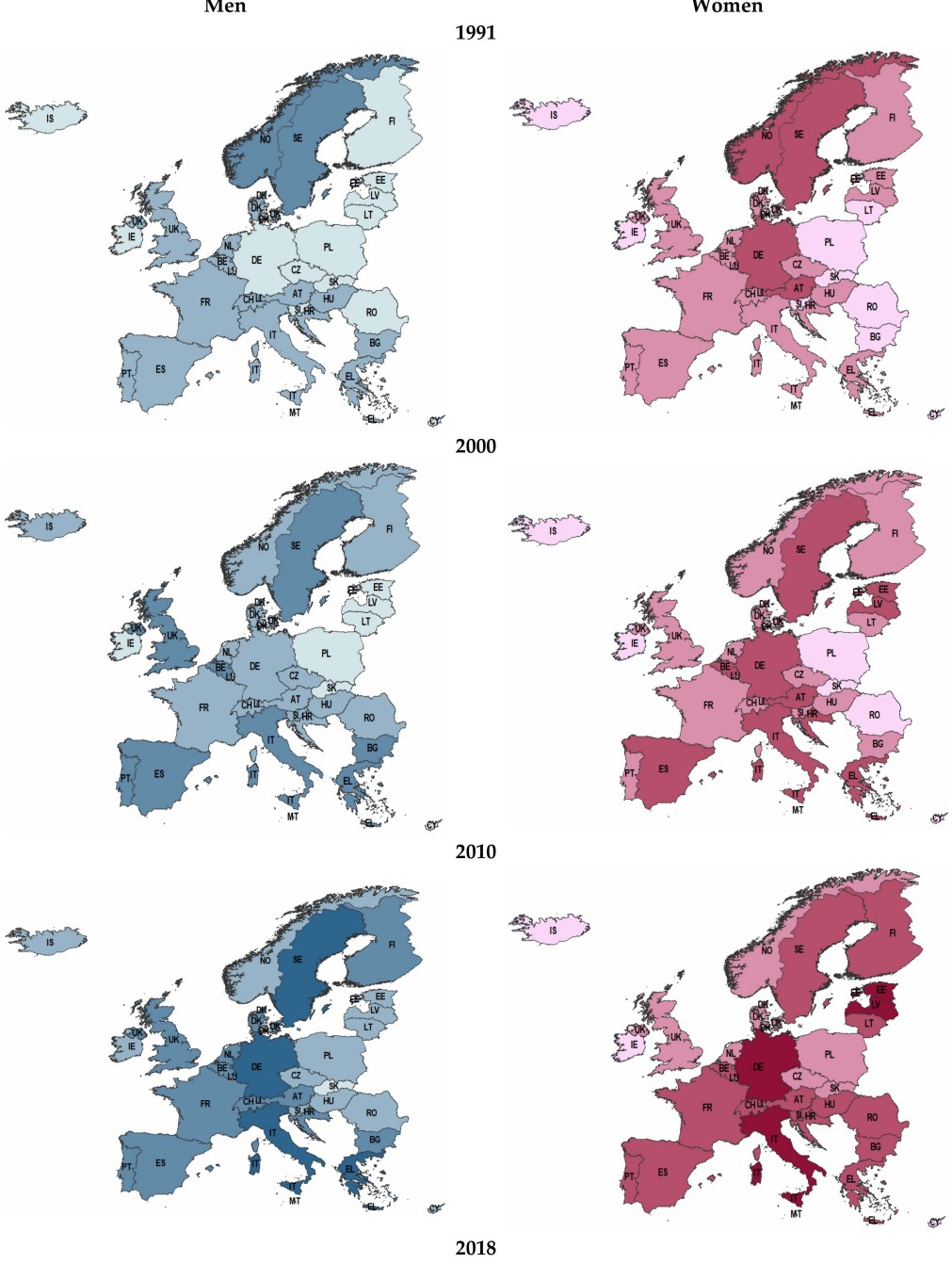

**Figure 5.** *Cont.*

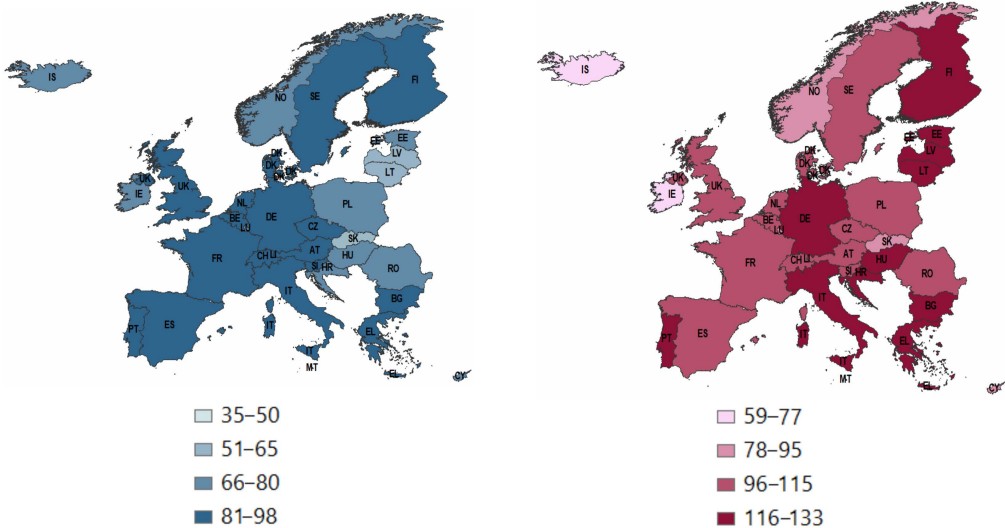

**Figure 5.** Numbers of women and men aged 65 and above per 1000 of the population in European countries in selected years.

### 3.2. Spatial Dynamic Shift–Share Analysis

The shift–share analysis was proposed by Creamer at the turn of the 1950s and 1960s [20]. Beginning in 1960, it has been popularized under the name of shift–share analysis (SSA) in research on the diversity of regional economic growth [21].

The method enables the multidimensional assessment of the dynamics of various processes. As a result of calculations, the so-called total net effect is determined, i.e., a relative change in the phenomenon in a region less a permanent total change. The net effect allows the indication of regional, local, structural, and sectoral factors determining the dynamics of examined processes. The classical (static) approach to the method defines the net change (effect) as follows:

$$tx_r. - tx.. = \sum_i u_{r\cdot(i)}(tx._i - tx_{..}) + \sum_i u_{r\cdot(i)}(tx_{ri} - tx_{.i}), \tag{1}$$

where $tx_r$.—growth rate (positive or negative) of the phenomenon in the current period in relation to the base period in the $r$ region (gross effect), $tx_{..}$—growth rate in the reference area, $u_{r\cdot(i)}$—regional weights (shares) for the $r$ region in the total variable value (reference value), $tx_r. - tx_{..}$—total net effect, pure net growth defined as a difference between the regional growth rate and the growth rate for the reference area, and $(tx._i - tx_{..})$—structural factor of regional growth, with $tx._i$ being the average growth rate of variable $x$ in the $i$ sector (section, category). The structural effect is equal to the weighted average of deviations between average growth rates in sectors and the growth rate in the reference area. It defines the degree of regional differentiation due to changes in the phenomenon structure; $(tx_{ri} - tx_{.i})$—geographical growth factor in the $i$ sector of the $r$ region for $tx_{ri}$ being the individual growth rate of variable x in the i sector and in the r region. That is a weighted average of regional deviations $(tx_{ri} - tx_{.i})$ assigning categories of the cross-sectional qualitative criterion to the respective regions (hence, that is the average effect of belonging to the $r$ region). Therefore, the geographical effect is also considered to be a local, specific effect and a regional competitiveness effect.

The static nature of the classical approach (1), on the one hand, is a simplification of the calculation process (changes are determined merely for two extreme study periods), but, on the other hand, it is its disadvantage. If the dynamics of the phenomenon is studied over many periods, it seems unfounded to assume the stability of the occurring phenomena in the meantime.

In 1988, Barff and Knight [22] modified the classical SSA method. The modification consisted in adopting time-varying regional weights and calculating recursively. That means that changes in the

value of the examined characteristic and individual effects are determined sequentially for each pair of consecutive periods. The results obtained are then added up:

$$\sum_z (tx_r . - tx_{..}) = \sum_z \sum_i u_r ._{(i)} (tx_{. i} - tx_{..}) + \sum_z \sum_i u_r ._{(i)} (tx_{ri} - tx_{.i}),$$ (2)

where *z*—periods of analysis.

However, classical and dynamic shift–share analysis approaches do not assimilate the idea of spatial relationships among regions. In consequence, examined objects are treated as economically and geographically unrelated areas, hence contradicting the assumption of Tobler's spatial autocorrelation theory [23]. In 2004, Nazara and Hewings [24] proposed introducing elements of the spatial weights matrix to the classical structural–geographical equation (SSSA, spatial shift–share analysis). One of the variants of the method, taking into account spatial relationships, is the following Equation (3):

$$tx_r . - tx_{..} = \sum_i u_r ._{(i)} (\mathbf{W}tx_{. i} - tx_{..}) + \sum_i u_r ._{(i)} (tx_{ri} - \mathbf{W}tx_{.i}),$$ (3)

where $(\mathbf{W}tx_{. i} - tx_{..})$—structural factor of regional growth, $(tx_{ri} - \mathbf{W}tx_{.i})$—local (geographical) growth factor in the *i* category (sector) of the *r* region, and **W**—adopted spatial weights matrix. In practice, it is assumed that spatial weights of the **W** matrix change or do not change over time. In the abovementioned publication by Nazar and Hewings, various possible SSSA variants were presented.

Population aging shows a long-term trend that began in Europe several decades ago (Figures 1–3). The trend is visible in the transformation of the age structure of the population and is reflected in a growing share of older people, along with a declining share of working-age individuals in the total population [4]. Therefore, in this study, due to the long-term and spatiotemporal changes occurring in the structure of the elderly age population, a dynamized SSA approach was applied:

$$\sum_z (tx_r . - tx_{..}) = \sum_z \sum_i u_r ._{(i)} (\mathbf{W}tx_{. i} - tx_{..}) + \sum_z \sum_i u_r ._{(i)} (tx_{ri} - \mathbf{W}tx_{.i}),$$ (4)

where $tx_r .$ means the rate of increase in the number of people aged 65 and above per 1000 of the population in the $z_n$ current period in relation to the previous (base) $z_i$ period in the *r* country; $tx_{..}$ is the rate of change in the phenomenon in Europe (being the reference area); $u_r ._{(i)}$ are regional weights for the *r* country in the form of shares of the analyzed variable in the total value of the variable in a given country for the *i* age category (men and women in this case, in five age groups); $(tx_r . - tx_{..})$ is the total global net effect, i.e., the difference between national and European growth rates in the number of people in the studied age group (not including the impact of neighborhood); and $(\mathbf{W}tx_{. i} - tx_{..})$ means the structural factor of growth rate equal to the deviation between the average and spatially weighted growth rates in the number of people aged 65 and above per 1000 of the population in *i* categories less the average growth rate. In the conducted analysis, negative values of structural indicators $(\mathbf{W}tx_{. i} - tx_{..})$ indicate that the pace of changes in the *i* age category is slower in locations (defined in matrix **W**) neighboring the *r* examined country. In turn, $(tx_{ri} - \mathbf{W}tx_{.i})$ is a geographical, differentiating, specific growth factor of people aged 65 and above per 1000 of the population in the *r* country, being the average effect of belonging to the *r* region. It has been mentioned above that the effect is also called the local competitiveness indicator, because it can be used to analyze the occurrence of internal changes in the *i* age category related to the competitiveness of the *r* country in relation to neighboring countries (according to the **W** matrix). What is more, it expresses the dynamics of changes in the age structure, specific to a given unit compared to the pace of changes in neighboring countries. Thus, a negative value of the indicator shows that the pace of aging of the *r* country in the examined periods is lower compared to that of the other countries, i.e., a given region is competitive in terms of the phenomenon growth rate in the *i* category relative to other regions (a positive value of the spatial–geographical effect is interpreted in the opposite way); **W**—spatial geographical distance matrix (more on the selection and construction of the matrix in Section 3.3). The model described by Equation (4) was applied for each pair of consecutive years. The results were added up.

### 3.3. Spatial Matrix—Reasoning and Selection

The choice of spatial weights matrix is a serious methodological problem [25,26]. Getis and Aldstadt [27] indicate general principles that should be followed when choosing a matrix. They emphasize that less complicated and easier to interpret matrices are preferred. Moreover, when analyzing spatial relationships, distances rather than just immediate vicinity should be taken into account. An important assumption regarding the selection of a spatial weights matrix was also formulated by LeSage [28]. He stated that the choice of a matrix depends on the nature of the problem and additional exogenous information not included in the data set. Therefore, knowing the local reality is extremely important, as interregional relationships may result from the spatial aggregation of causes rather than diffusion of effects.

Data presented in Figures 4 and 5 show that European countries were grouped into homogeneous, compact areas of population aging in the years 1991–2018. Wulff and Ejlskov [29], Suchecka and Urbaniak [30], Sundström et al. [31], and Wiktorowicz [32] concluded that the regionalized aging of the European population may be associated with a certain tendency towards spatial concentration of the process determinants.

Taking into account the above assumptions, the specificity of the analyzed phenomenon, and maintaining the above-mentioned condition of exogeneity [33], this analysis applies spatial weights matrix **W** based on geographical distance, where $w_{rk} = 1$ when the distance between the $r$ and $k$ countries is smaller than a certain set radius g, and $w_{rk} = 0$ for the distance between the $r$ and $k$ countries greater than $g$. In this analysis, $g = 1437$ km, because it is the smallest physical distance from the geographical centers of the countries determined in such a way so that each country has at least one neighbor. The element of the structural and geographical equation in model (4) takes the form:

$$\mathbf{W}tx._i = \frac{\left(\sum_{k=1}^{R} w_{rk} X_{ki}^* - \sum_{k=1}^{R} w_{rk} X_{ki}\right)}{\sum_{k=1}^{R} w_{rk} X_{ki}}, \tag{5}$$

where for $i = 1, \ldots , S$, $r, k = 1, \ldots , R$, $X_{ki}$ is the number of people aged 65 and above per 1000 of the population in the $k$ country for the $i$ age group, $X_{ki}^*$ is the same value for the final period, and $w_{rk}$ is an element of the spatial weights matrix for the $r$ and $k$ countries. It was assumed that spatial weights did not change over time, and matrix elements were standardized by rows to 1.

## 4. Results and Discussion

### 4.1. Interpretation of SSSA Components

In the years 1991–2018, the number of people aged 65 and above per 1000 of the population in the 32 European analyzed countries increased in total by 37%. There was no country in which the number of elderly people fell (Table 1, Figures 4–6).

The population was aging the fastest in Malta, where the population aged 65 and above per 1 thousand inhabitants increased by 66% in 2018 compared to 1991 (an average annual rise in the population in the studied age group was 2.5%). The growth rate of the elderly population in Malta was 29 percentage points faster than the average growth rate of the phenomenon in Europe (the European net effect = 37%, Figure 6). Countries where the above 65 population growth rate was equally high were Slovenia and Liechtenstein. In the indicated countries, there was an increase of about 60% in the population aged 65+ per 1 thousand inhabitants (with an average annual growth rate of around 2%). Poland also belonged to the group of countries characterized by the fastest aging of the population (an absolute increase in the population aged 65 and above reached the level of 52.7% in the analyzed period). The Polish population was aging, on average, by 16 percentage points faster than the European one, Figure 6.

The slowest aging countries were Luxembourg (a growth rate of 10.4% and an average annual growth rate of 0.4%) and Norway (NO $tx_r$ = 5.4%, an average annual growth rate of 0.2%), Table 1. In Norway, the growth rate was by 32 percentage points slower than the average in Europe and by about 62 percentage points slower than in Malta, Figure 6.

**Table 1.** Dynamics of population aging in European countries in 1991–2018 (growth rate in %).

| GROWTH RATE OF AGING ($tx_r$) | COUNTRIES |
|---|---|
| The fastest for $tx_r \in$ <45.8; 66) | Portugal (45.8), Greece (46), Finland (47.2), Latvia (52), Estonia (52.3), POLAND (52.7), Romania (54.1), Lithuania (55.9), Liechtenstein (59.2), Slovenia (59.3), Malta (66) |
| Fast for $tx_r \in$ <25.6; 45.8) | Cyprus (44.2), Bulgaria (44.2), the Czech Republic (43), Slovakia (41), Italy (40.4), the Netherlands (39.6), Germany (37), France (34.8), Spain (34.2), Hungary (33.9), Croatia (33.3), Iceland (31.9) |
| Moderate for $tx_r \in$ <10.8; 25.6) | Sweden (13.6), the United Kingdom (16.1), Ireland (21.6), Belgium (22.7), Denmark (22.9), Austria (23.6), Switzerland (24.4) |
| The slowest for $tx_r \in$ <5.4; 10.8) | Norway (5.4) and Luxembourg (10.4) |
| EUROPEAN GROWTH RATE ($tx_{..}$): 37 | |

Note: the groups were created based on the formula included in: Kukuła (2004, p. 28).

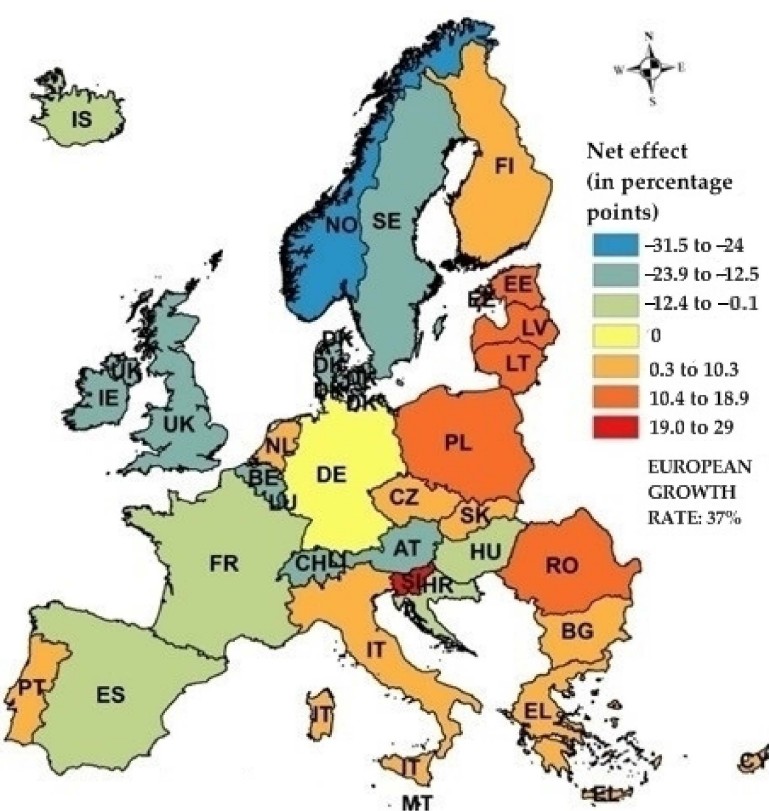

**Figure 6.** Net effect of the number of individuals aged 65 or over per 1000 of the population in Europe (in percentage points).

In the years 1991–2018, an increase in the number of women and men per 1 thousand of the population was observed in all the analyzed age groups: 65–69, 70–74, 75–79, 80–84, and 85 and above, Figure 7. Nevertheless, men aged 85 and older were characterized by the most dynamic population aging. During the study period, the number of men in that age group increased by more than 96%, whilst the growth rate was by about 60 percentage points higher than the average in Europe (an average annual growth rate for men was about 4%). The slowest growth rate was recorded in the group of men

aged 65–69 and women aged 75–79 per 1000 of the population. The increases were slower than the average growth rate by 28.3 and 25.4 percentage points, respectively, Figure 7.

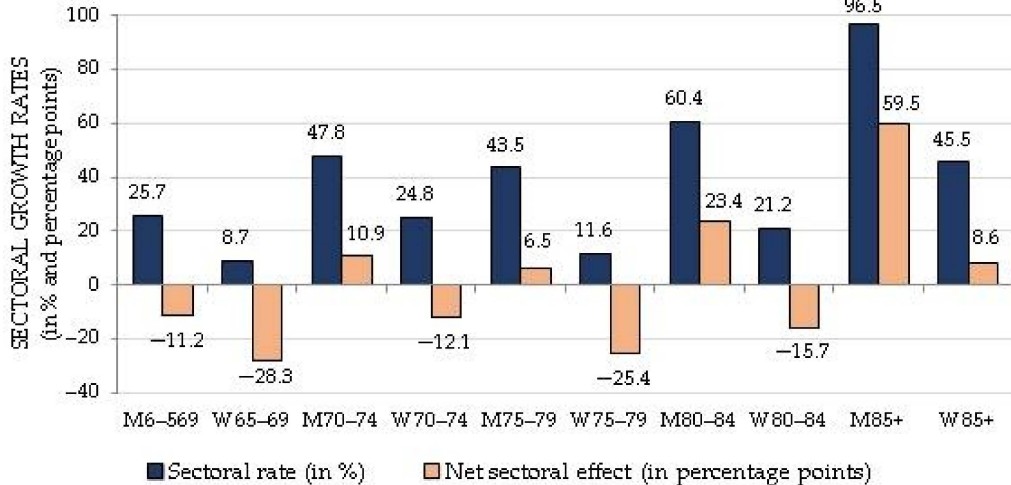

**Figure 7.** Sectoral growth rates by gender and specific age category of the number of people aged 65 and above per 1000 of the population (in % and percentage points). Note: M—men; W—women.

The pace of the European population aging varied, and the results of the conducted analysis indicate that the dynamics of changes in the number of people aged 65+ per 1000 of the population in individual countries (the net effect) was determined by changes in the population structure in specific age groups in neighboring regions (structural effects) and the local competitiveness of a given country relative to other units (geographical effects), Table 2.

**Table 2.** Values of spatial shift–share analysis (SSSA) components (in percentage points, pp).

| Country | NEF | SEF | GEF | Country | NEF | SEF | GEF |
|---------|-----|-----|-----|---------|-----|-----|-----|
| NO | −31.5 | −2.7 | −28.8 | IT | 3.4 | −8.5 | 11.9 |
| LU | −26.5 | −9.2 | −17.4 | SK | 4 | −10.6 | 14.6 |
| SE | −23.4 | −3 | −20.4 | CZ | 6 | −10.1 | 16.1 |
| UK | −20.9 | −5 | −15.8 | BG | 7.2 | −9.9 | 17.1 |
| IE | −15.3 | −5.9 | −9.5 | CY | 7.2 | −6.7 | 13.9 |
| BE | −14.3 | −7.4 | −6.9 | PT | 8.8 | −7.8 | 16.5 |
| DK | −14 | −5 | −9 | EL | 9 | −8 | 17 |
| AT | −13.3 | −8.5 | −4.8 | FI | 10.3 | −7.5 | 17.8 |
| CH | −12.5 | −5.9 | −6.6 | LV | 15 | −14.2 | 29.2 |
| IS | −5 | −4.4 | −0.6 | EE | 15.4 | −13.6 | 29 |
| HR | −3.6 | −10.1 | 6.5 | PL | 15.7 | −11.2 | 26.8 |
| HU | −3 | −10.1 | 7.1 | RO | 17.2 | −11.8 | 28.9 |
| ES | −2.8 | −6.2 | 3.4 | LT | 18.9 | −12.5 | 31.4 |
| FR | −2.1 | −6.7 | 4.5 | LI | 22.3 | −6.7 | 28.9 |
| DE | 0.004 | −9.68 | 9.69 | SI | 22.4 | −12.1 | 34.5 |
| NL | 2.6 | −6.3 | 8.9 | MT | 29 | −6.5 | 35.6 |

Note: NEF—netto effect; SEF—structural effect; GEF—geographical effect.

Based on the SSSA results contained in Table 2, it was observed that the estimated structural effects were negative in all the studied countries. Therefore, it can be concluded that the population growth rates in individual age groups in a given country were corrected for negative structural effects, i.e., slower changes in the structure of the population aged 65+ per 1000 thousand inhabitants in neighboring units in the years 1991–2018, whereas a key factor differentiating the pace of elderly population growth was geographical effects. Competitiveness indicators took negative and positive values, thus polarizing countries in terms of the population aging dynamics in individual age groups.

Positive effects characterized socially disadvantaged, i.e., faster aging, countries. The above-average positive value of the geographical effect ($\overline{GEF}$ = 18.6) showed a country's growing uncompetitive position in relation to the defined neighboring countries in the analyzed period. The group included, among others, Malta, Slovenia, Lithuania, Latvia, Estonia, Romania, Liechtenstein, and Poland. Developing countries predominated in the group. For example, in Malta, where the fastest relative increase in the older age population per 1000 inhabitants was recorded ($NEF_{MT}$ = + 29 pp.), the value of the total effect was more strongly influenced by the positive value of the geographical effect ($GEF_{MT}$ = 35.6 pp). The situation indicates the country's uncompetitiveness, resulting from a specific, higher dynamics of 65+ population growth (despite the fact that the net effect value in Malta was affected by the slower population aging process in neighboring countries—an adjustment of the growth rate for the value in individual groups caused by the impact of structural changes in neighboring countries was at the level of $SEF_{MT}$ = −6.5 pp).

In turn, the above-average negative value of the geographical effect ($\overline{GEF}$ = −12) reflected a high competitive position of a country in relation to neighboring units. An exceeding value of the above effect also indicated an impact of specific factors and local conditions on the phenomenon growth rate greater than that of changes in the age structure of the elderly population. Norway, Sweden, Luxembourg, and the United Kingdom belonged to the group of "competitive" countries with a much slower rate of population aging. Thus, in Norway, the net effect ($NEF_{NO}$ = −31.5 pp) was mainly influenced by the negative value of the geographical effect ($GEF_{NO}$ = −28.8 pp), and the value indicated the local competitiveness of the country (in relation to neighboring countries, defined in matrix W) in terms of the population aging pace. Therefore, there was a specific, slower dynamics of changes in the structure of the population aged 65+ per 1 thousand inhabitants in Norway, which in turn might be conditioned by local socioeconomic factors. The results of the analysis also indicate a slight impact of slower structural changes in neighboring regions on the population growth rate in Norway ($SEF_{NO}$ = −2.7 pp).

The results of the carried out analysis also identified countries where the negative value of the net effect (i.e., slower population aging than the average in Europe) was influenced by the dynamics of changes in the age structure of the population in neighboring countries (defined in matrix W) more than by local competitiveness (the absolute value of structural effect was negative and exceeded the geographical effect value). That was the case in France, Spain, Hungary, Croatia, and Austria (where the values of both effects were negative), Table 2.

*4.2. Discussion of Potential Determinants of SSSA Results*

Although the SSSA method did not indicate the direct, socioeconomic causes of the phenomenon growth rate, the careful analysis of the net effect components, i.e., structural and geographical effects, enabled identifying age categories in which the population growth rate significantly affected the aging rate in a given country, Table 3.

**Table 3.** Structural and geographical effect components impacting on the 65+ population growth rate in selected countries (in percentage points, pp).

| Country | NEF | SEF | 65–69 | | 70–74 | | 75–79 | | 80–84 | | 85+ | |
|---|---|---|---|---|---|---|---|---|---|---|---|---|
| | | | M | W | M | W | M | W | M | W | M | W |
| NO | −31.55 | −2.70 | −1.29 | −3.88 | 1.40 | −1.14 | 0.75 | −2.66 | 1.60 | −1.07 | 2.54 | 1.04 |
| LU | −26.53 | −9.15 | −1.64 | −4.93 | 0.91 | −1.82 | 0.19 | −3.44 | 0.98 | −1.74 | 1.69 | 0.64 |
| PL | 15.69 | −11.15 | −1.78 | −5.83 | 0.88 | −2.39 | 0.33 | −3.56 | 0.85 | −1.69 | 1.43 | 0.62 |
| AT | −13.33 | −8.49 | −1.56 | −4.89 | 0.71 | −1.93 | 0.41 | −3.25 | 1.04 | −1.62 | 1.88 | 0.72 |
| SI | 22.36 | −12.10 | −1.75 | −5.89 | 0.67 | −2.38 | 0.11 | −3.65 | 0.69 | −1.84 | 1.33 | 0.62 |
| MT | 29.03 | −6.54 | −1.58 | −5.14 | 1.41 | −1.70 | 0.69 | −2.72 | 1.25 | −1.11 | 1.79 | 0.57 |

**Table 3.** *Cont.*

| Country | NEF | GEF | 65–69 | | 70–74 | | 75–79 | | 80–84 | | 85+ | |
|---|---|---|---|---|---|---|---|---|---|---|---|---|
| | | | M | W | M | W | M | W | M | W | M | W |
| NO | −31.55 | −28.85 | −1.35 | −1.19 | −3.13 | −4.03 | −3.80 | −4.14 | −3.19 | −3.91 | −2.09 | −2.02 |
| LU | −26.53 | −17.38 | 0.27 | −3.16 | −1.49 | −3.59 | −2.15 | −3.23 | −0.99 | −2.16 | −0.27 | −0.62 |
| PL | 15.69 | 26.84 | 4.64 | 6.37 | 1.40 | 4.39 | −0.11 | 2.84 | 0.03 | 3.02 | 0.39 | 3.87 |
| AT | −13.33 | −4.83 | −0.16 | −3.92 | 1.05 | −0.73 | 1.66 | 0.43 | −1.03 | −2.39 | −0.27 | 0.52 |
| SI | 22.36 | 34.45 | 6.63 | 2.73 | 4.51 | 5.77 | 1.95 | 3.42 | 1.01 | 3.10 | 0.72 | 4.61 |
| MT | 29.03 | 35.57 | 5.59 | 6.70 | 4.73 | 7.27 | 0.74 | 3.79 | 0.42 | 3.41 | −0.01 | 2.92 |

Note: grey color—the highest pace of change (+ or −) in the age categories that significantly affected the net effect components in a given country.

The data contained in Table 3 show that the negative value of the geographical effect in Norway was conditioned by a slower pace of elderly population growth per 1000 inhabitants in all age categories in relation to neighboring countries (specified in matrix W). The results of the analysis indicate that Norway's slower aging rate was determined by a slower increase in the number of women aged 70–74, as well as women and men aged 75–79 compared to that in the other studied countries ($GEF_{W70-74}$ = −4.03 pp, $GEF_{M75-79}$ = −3.80 pp, $GEF_{W75-79}$ = −4.14 pp). On the other hand, in Luxembourg (ranking the second as the slowest aging country), the geographical effect was influenced by a slower growth rate in the number of women aged 65–69, 70–74, and 75–79 than that in neighboring countries ($GEF_{W65-69}$ = −3.16 pp, $GEF_{W70-74}$ = −3.59 pp, $GEF_{W75-79}$ = −3.23 pp). The specialist literature indicates that Norway has been carrying out one of the most effective and sustainable social policies aimed at old and aging people for years [34].

Norway's population is aging, as are the populations of other countries in Europe. The impact of aging in Norway is lessened by high fertility rates and positive net immigration. Nevertheless, policy and political responses to this aging population include pension reforms implemented on a phased-in timeline beginning in 2011, a plan that began in 2001 among the major political parties to develop an inclusive workplace policy, and a transformation of the national health care system designed to support the health and care needs of an older population, which includes more attention to primary and preventive care, support for independent living, and provisions for care delivered in the home. The government of Norway has been working to implement a strategy to create an 'age-friendly' society since 2015 [35]. The UN's Sustainable Development Goals (SDGs) and Agenda 2030 have been used as input to Norway's strategy [36]. Studies of elderly patients across Scandinavia show that chronic disease self-reports fell between 2002 and 2015 [37].

That same study revealed that people aged 65 and older reported better health and function over that time period, perhaps due to improvements in the support systems and environment, rather than their individual efforts [38]. Over 40% of the elderly (<67 years) reported that they were satisfied with their life in Norway, according to a study conducted by [39].

Norway has prioritized both equality across generations and support for issues facing aging people [33]:

- The labor force participation of older people is high and rising;
- Most older people are active in civic organizations, volunteering, family activities, and have regular social contact;
- Norwegian health policy is designed to emphasize individual empowerment and coping skills;
- That same health policy emphasizes health promotion and disease prevention, tailored to all life stages;
- Public health policy stresses active aging.

By contrast, Luxembourg faces a different situation regarding an aging population. While nearby countries have rectangular age "pyramids", indicating equal numbers of people in each age bracket, the demographics of the population in Luxembourg are skewed by a massive number of foreigners,

who tend to be much younger than citizens [40]. Moreover, Luxembourg follows a population aging trend that is expected to accelerate, not only because of a decline in mortality and fertility rates, but also because of people living longer [41]. The country might also have one of the highest life expectancies in Europe, at around 84 years, by 2040–2045 [42]. However, the Luxembourg healthcare system is one of the most comprehensive ones in the world, offering virtually unrestricted access to the Luxembourg population [43].

In turn, the aging of Malta's population was conditioned by the largest increase in the number of men aged 70–74 in Europe ($GEF_{M70-74}$ = 7.27 pp) and aged 65–69 ($GE_{M65-69}$ = 6.70 pp). In Malta, as in Norway, local factors exerted the greatest impact on the growth rate of the phenomenon in the country ($NEF$ = 29.03 pp, $SEF$ = −6.54 pp, $GEF$ = 35.57 pp). Nevertheless, in the case of Malta, the indicated geographical effect components determined the country's least favorable position in the ranking of the fastest-aging European countries. Malta is an example of a country suffering from inversion of the age pyramid, according to Formosa's findings [44]. The population of the Maltese islands has shifted due to decreased fertility and increased life expectancy. Per capita expenditure on health care for people over 65 (men) and 68 (women) is high and increases sharply for people older than 75.

The increased costs of social protection for elderly citizens may result in aging being a burden, rather than an achievement [45]. The Maltese government has been working to implement the concepts of active aging promoted by the World Bank, the WHO, and the EU since 2011, but it is a challenging effort. Policies adopted before the turn of the century require updating, modernization, and revision to be sustainable considering the current population trends, with the intent of encouraging active participation and involvement of older people [44]:

- Increasing social awareness of the benefits of involvement of and involving older people in community activities;
- Addressing barriers to involvement, particularly for older people who are not able to live independently or who are frail;
- Considering the needs and desires of an older population for involvement;
- Providing opportunities and increasing motivation to participate.

By contrast, Poland and Slovenia belonged to the group of countries where the population aging rate was definitely higher than the European growth rate of the studied phenomenon ($NEF_{PL}$ = 15.69 pp, $NEF_{SI}$ = 22.36 pp). The study results indicate that faster aging of the population in the abovementioned countries was conditioned by the specific dynamics of population changes in individual age groups, compared to the pace of sectoral changes in neighboring units (geographical effects were positive and outweighed structural effects). The data contained in Table 3 also show that the geographical effect value in both countries was most strongly influenced by, among others, a rapid increase in the number of women aged 85 and above ($GEF_{PLW85+}$ = 3.87 pp, $GEF_{SIW85+}$ = 4.61 pp). As can be seen, a common feature of countries aging faster on average was also a faster increase in the number of men aged 65–69 and women aged 70–74 and 80–84. Demographers describe that process as double aging. At first, an increase in the general number of older people in a society is observed. Then, within that group, there is an increase in the number of people of advanced age, who are called old-old (the percentage of individuals aged 80 years and above is higher) [46]. This takes place both in Poland and Slovenia. Moreover, Poland belongs to a small group of European countries where a total of three phenomena have recently occurred, exerting a major impact on demographic dynamics: decreased fertility, increased emigration, and lack of immigration [47]. What seems particularly important for Poland is the long-term nature of those processes and their convergence over a period of at least 25 years. Owing to fewer births than in the past and rising life expectancy, Slovenia also faces the process of population aging, which is expected to be faster than in the other European countries, according to demographic projections [48]. Furthermore, the aging policy in Slovenia severely suffered from the effects of the economic crisis [49]. In order to comprehensively address those challenges, Slovenia's and Poland's governments have adopted the Active Aging Strategy, the long-term care

system [50], as well as assessed the chances to reduce unemployment among people before retirement age and thus decrease government spending [51].

A separate group of countries were those where the growth rate was slower than the average in Europe and conditioned by changes in the age structure of the subjects. For example, in Austria, there was a faster increase in the number of women aged 65–69 than in neighboring units ($SEF_{ATW65-69}$ = −4.89 pp), but a slower increase in the population aged 80–84 and 85 years and above ($SEF_{ATM80-84}$ = 1.04 pp, $GEF_{ATW80-84}$ = −2.39 pp, $SEF_{ATM85+}$ = 1.88 pp, $SEF_{ATW85+}$ = 0.72 pp). In Austria (as one of the industrialized societies), the aging of the population was influenced by low fertility, increasing life expectancy, and moderate migration. Despite the 2000s reforms aimed at closing off early retirement paths, boosting labor market participation and delaying retirement transition, the aging process has become more apparent [52]. Within 30 years, the balance of age groups in Austria is expected to shift to half being older than 48.3 years and one-third being at least 65 years old. The median age of the population is expected to rise to 48.3 from 38.2 by 2050 [53]. The Austrian government has responded to this finding by introducing a federal plan in 2011 that set a direction for support of active aging and better quality of life. However, this plan provided guidelines only rather than specific political actions [54].

Population aging affects all European countries; hence, it has become one of the main issues in international discussion on balanced growth and integrated development.

## 5. Conclusions

The aim of the article was to analyze the dynamics of the aging process in Europe. Based on the conducted research, it was found that there was an increase in the population aged 65 and above per 1000 inhabitants in each of the analyzed countries in the years 1991–2018 (the total increase was 37% in the studied countries). The average annual rise in the number of elderly people was 1.4% (two people). In addition, the number of senior citizens was characterized by quite large spatial diversity; a clear division of continental Europe into the older western part and the younger eastern one could be seen. Throughout the analyzed period, women clearly predominated among the elderly; the number of women aged 65 and above per 1000 of the population was also more spatially diversified than the number of men in Europe.

The populations of Malta, Slovenia, Liechtenstein, Lithuania, and Romania were aging the fastest. Poland, like the abovementioned European countries, was struggling with the dynamic aging problem as a result, on the one hand, of extended population longevity and, on the other hand, a decreased number of births. The slowest (slower than the European growth rate) aging processes were recorded in Norway and Luxembourg.

The results of the performed analysis show that the dynamics of population aging in European countries was determined by changes in the age structure of women and men (structural and sectoral effects), spatial relationships, and specific, slower or faster dynamics of changes in the structure of the population aged 65+ per 1000 inhabitants (geographical effects). The male population was aging much faster than the female one. The most dynamic population aging process was characteristic of men aged 85 and above (it was over 50 pp faster than the aging process for women in that age group). The slowest increase in the number of people was recorded in the groups of women aged 65–69 and 75–79. Nevertheless, the key factor differentiating the pace of advanced age population growth was not the dynamics of changes in the age structure (also in neighboring countries), but geographical effects (called local competitiveness factors).

The SSSA does not provide an answer to the question about specific local factors and consequences of a phenomenal growth rate in the senior population but does allow for a faster and more comprehensive understanding of the analyzed process. Our findings supported a comprehensive study of the dynamics of population aging, an indication of the causes of that process and identification of benchmark countries—Norway, Malta, and Austria—with the greatest achievements in terms of implementing old age policy. The results, combined with the knowledge of the locally implemented sustainable

social policies aimed at the elderly and old age, can enrich the conclusions about the conditions and directions of European countries' aging populations and hence may be of interest to policymakers and service providers.

Moreover, by introducing spatial elements to the shift–share model, this study conducts research into the aging structure of European countries by considering the regional aging spatial effect, which extends the research scope of the original shift–share model. The national growth rate calculated in a classical way, i.e., without taking into account the dynamics in the age structure occurring in intermediate years of the examined period, and without considering the fact that no country appears as a separate geographical area, would be 44%. That would mean an increase of 44% in the number of people aged 65 and above per 1000 of the population in 2018 in relation to 1991 (by 7 pp more than the results obtained employing the spatial dynamic method). In general, the values of effects determined using the classical method are overstated as the SSSA method takes into account the presence of a specific dynamics of changes compared to neighboring countries and introduces an adjustment for growth in individual age groups (including spatial interactions) caused by the growth rate of neighboring countries.

Undoubtedly, the values of the received results were influenced by the nature of the adopted spatial weights matrix. The issue of matrix selection, an in-depth explanation of the causes and consequences of the European society aging, taking into consideration the spatial distribution of the phenomenon and the local conditions of each unit (country), are the main directions for further research by the authors of this publication.

**Author Contributions:** Conceptualization, E.A. and K.L.-G.; Formal analysis, E.A. and K.L.-G.; Investigation, E.A. and K.L.-G.; Methodology, E.A.; Validation, E.A. and K.L.-G.; Visualization, E.A. and K.L.-G.; Writing—Original Draft, E.A. and K.L.-G.; Writing—Review and Editing, E.A. and K.L.-G.

**Funding:** This research received no external funding.

**Conflicts of Interest:** The authors declare no conflict of interest.

## Appendix A

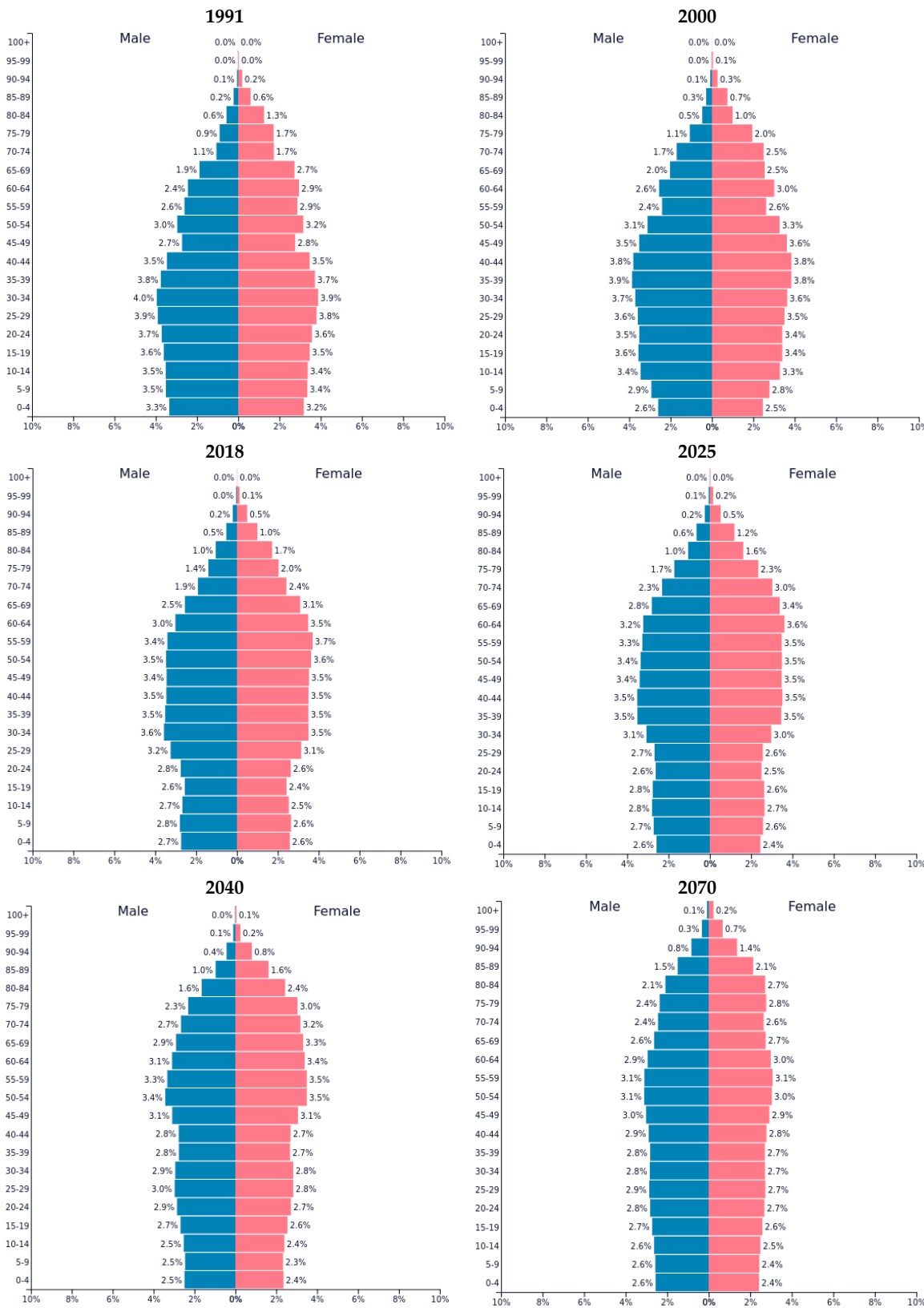

**Figure A1.** Population pyramids of Europe. Note: The source of the charts is PopulationPyramid.net [55].

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
