# Peer review of "How Fast Is Europe Getting Old? Analysis of Dynamics Applying the Spatial Shift–Share Approach"

_sustainability, doi:10.3390/su11205661_

Round 1
Reviewer 1 Report
The article addresses a topic of interest in the field of geodemography, but the presentation of the scientific concept should be improved. The article should contain a much more applied part about the consequences of demographic aging in Europe. It would be useful to make some pyramids of the structure by age groups for the analyzed period and their projection for the future.
Author Response
We appreciate the Reviewer’s decision and we would like to thank for all the comments and remarks. Please, find below point-by-point responses to each of the suggestions as well as our own comments.
The article should contain a much more applied part about the consequences of demographic aging in Europe.
Please find some information in lines: 37-40, as well as all in lines: 41-92, where of the SD goals widely discuss the consequences and remedies on ageing process in Europe. Consequences are also discussed in the second chapter – lines 130-157. Moreover, we do see the above mentioned issues, as the great research opportunities to be solved in the future and this is also suggested in the Discussion of the paper.
It would be useful to make some pyramids of the structure by age groups for the analyzed period and their projection for the future.
In accordance with the Referee’s recommendation, the population pyramids that show changes in the age structure for selected years and its forecasts have been added in Appendix A and introduced in text – lines 29-31.
Reviewer 2 Report
Major comments:
The paper needs a full discussion (not just two or three brief references as it is now in the current version) on the effects that international migrations within Europe and between Europe and other areas produce on the ageing process. Nothing is said about the ageing process in other areas such as the USA, China or LDCs. For the average reader it is interesting to know, briefly stated, whether the European behavior is applicable to these zones. In Conclusions a reference to the following policy implication is missed: Is possible or necessary, attending to the results of the paper, a single and common Ageing Policy for all the EU? (or, due to the specific characteristics of the countries involved, this is not a recommendable target).
Minor comments:
In Table 1, the two thresholds 25.6 for “Fast” and “Moderate”, are they correct? In Figure 6, 1 cm does not represent 229 km, as stated. Line 408. I think Austria is not the same case as France, Spain, Hungary and Croatia. These four countries show negative SEF, larger in absolute value than a positive GEF (Table 2). Austria has both effects negative. Table 3. The explication for the grey colours (“the highest pace of change (+ or -) in the country”) is not convincing.
Author Response
We corrected the paper according to the Reviewer’s remarks. Comments were highly insightful and enabled us to greatly improve the quality of the manuscript. Please, find below point-by-point responses to each of the suggestion as well as our own comments.
The paper needs a full discussion (not just two or three brief references as it is now in the current version) on the effects that international migrations within Europe and between Europe and other areas produce on the ageing process
In accordance with the Referee’s recommendation, the discussion on the impact of migration on population ageing has been added in the chapter 2.
Nothing is said about the ageing process in other areas such as the USA, China or LDCs. For the average reader it is interesting to know, briefly stated, whether the European behavior is applicable to these zones.
In accordance with the Referee’s recommendation the information on the aging of the population in other parts of the world and the position of Europe against them has been added in the introduction – lines 29-34.
In Conclusions a reference to the following policy implication is missed: Is possible or necessary, attending to the results of the paper, a single and common Ageing Policy for all the EU? (or, due to the specific characteristics of the countries involved, this is not a recommendable target).
We agree with the Referee and we have added the paragraph in lines 568-576 and 538-539.
Table 1, the two thresholds 25.6 for “Fast” and “Moderate”, are they correct?
In accordance with the Referee’s recommendation we have corrected the mistake as we moved the countries with Moderate pace of change to the group of regions with Fast rate of change in the number of people aged 65 or over per 1000.
Figure 6, 1 cm does not represent 229 km, as stated.
In accordance with the Referee’s recommendation we have deleted the note about the scale (when the size of the Figure changes the scale becomes not proper as well).
Line 408. I think Austria is not the same case as France, Spain, Hungary and Croatia. These four countries show negative SEF, larger in absolute value than a positive GEF (Table 2). Austria has both effects negative.
Austria (together with the France, Spain, Hungary and Croatia) is in the case of the country where the negative value of the net effect was influenced by the dynamics of changes in the age structure of the population in neighbouring countries more than by local competitiveness. The SEF effect value was negative and exceeded the geographical effect value. However, we have added a relevant explanation in line 412.
Table 3. The explication for the grey colours (“the highest pace of change (+ or -) in the country”) is not convincing.
We agree with the Referee and we have corrected the explanation on the grays in the Table 3.